# Finite Element Modeling and Experimental Investigation for Wood/PVC Composites Log-Walls under In-Plane Lateral Load

**DOI:** 10.3390/polym14214673

**Published:** 2022-11-02

**Authors:** Warawit Eakintumas, Tawich Pulngern, Vichai Rosarpitak, Narongrit Sombatsompop

**Affiliations:** 1Department of Civil Engineering, Faculty of Engineering, King Mongkut’s University of Technology Thonburi, Bangmod, Thongkru, Bangkok 10140, Thailand; 2V.P. Wood Co., Ltd., Suksawat 41, Banphueng, Phra Pradaeng, Samut Prakan 10130, Thailand; 3Polymer PROcessing and Flow (P-PROF) Research Group, Division of Materials Technology, School of Energy, Environment and Materials, King Mongkut’s University of Technology Thonburi, Bangmod, Thongkru, Bangkok 10140, Thailand

**Keywords:** wood polyvinyl chloride composites, log-house, load-bearing wall, finite element model, full-scale experiment

## Abstract

This work experimentally determines the in-plane lateral load behavior of a full-scale WPVC composite log-wall, with and without additional through-bolts. The results indicate that the WPVC composite log-wall panel with through-bolts produced higher hysteretic parameter values in terms of strength and energy dissipation than the log-wall without through bolts due to a reduction in wall uplift (48.2% for secant stiffness of cycle, 39.5% for hysteretic energy at the last displacement level). The WPVC composite log-wall panel with through-bolts presented better structural stability and was recommended for investigation. A finite element model (FEM) of a WPVC composite log-wall panel with through-bolts was created using beam elements as log-members and multilinear plastic links as connections, and was verified by the experimental results. The verified FEM was used for further parametric study of wall dimensions and first log-foundation locations. The parametric investigations indicated that increasing panel height and width unfavorably affected lateral load capacity, monotonic and cyclic stiffness, and energy dissipation. The cyclic stiffness decreased by 39% while energy dissipation increased by 78.8%, for the last displacement level when the wall height was increased from 2.350 m to 3.525 m. The cyclic stiffness and energy dissipation of a panel with a width of 6 m decreased 14% and 24.4% compared to a panel with a width of 3.5 m. Moreover, moving log-foundation connections from the original position to the edges of the panel improved performance under monotonic and cyclic horizontal loads; an increase in the number of log-foundation connections had an insignificant effect on panel behavior.

## 1. Introduction

A log-house is a traditional wooden house that can be erected without heavy machinery in construction. The log-house wall (log-wall) is a load-bearing wall structure that consists of log-element series. Log-elements of main and orthogonal walls are stacked horizontally upon each other to interlock at the corners. As the log-wall is the main structural component resisting external vertical and horizontal loads, many research studies have focused on log-wall behavior under vertical [1,2,3,4,5] and lateral loads [6,7,8,9,10,11,12,13,14,15,16,17,18,19,20,21], which had the details are as follows.

In the last three decades, Heimeshoff and Kneidl [1] have investigated lab-scale and full-scale solid timber log-wall panels with different opening sizes and proposed an analytical model [2] to estimate acceptable vertical load. In 2015, the buckling loads of glue-laminated timber log-wall structures with different geometries under in-plane compression were presented experimentally by Bedon et al. [3] and compared with the classical theory of column-buckling and plate-buckling analytical solutions. Bedon et al. [4,5] used a finite element model (FEM) to predict the buckling load and proposed buckling design curves using Eurocode 5. The structural behaviors of a solid timber log-wall structure under lateral load were assessed using different methods (full-scale wall experiment [6,7,8,9,10,11,12,13,14], full-scale shaking table test [15,16,17], 3D solid FEM [13,18,19], simplified FEM [12,14,17,20,21,22,23]).

In 2002, Gorman and Shrestha [6,7] evaluated the lateral load resistance of handcrafted timber log-walls. According to the Structural Engineers Association of California (SEAOC), the cyclic displacement protocol was implemented on the panels with five different connector configurations (location and number of threaded rods, corner intersection, and inserting galvanized pipe as threaded rod sleeve). This research was further studied by Popovski [8]. The metal connectors were replaced with hard and soft wood pins. Two quasi-static tests (pushover and cyclic) were conducted on the full-scale timber log-wall.

To understand the actual lateral load (quasi-static cyclic load) resistance of a log-wall structure, Hirai et al. [9] studied the combination of lateral resisting elements for a log-house system, which consisted of vertical through-bolts, dowels between log-layers, interlocking between main and cross logs (corner joint), and the log-layer friction. The experiments indicated that the lateral load resistance of the timber log-wall was the superposition of individual lateral resisting-element capacities and that the vertical through-bolt should be considered as an inclinational resisting-element instead of a lateral resisting-element.

In 2006, Yeh et al. [10] used lag screws as mechanical fasteners between log-elements and studied the effect of openings on the monotonic lateral load resistance of a D-log-wall. The results suggested that additional lag screws around the opening increased the horizontal shear strength of the panels. The influences of aspect ratio (height per width) and seismic response parameters of a timber log-wall fastened with lag screws under cyclic loading were investigated by Graham et al. [11]. The results can be used to estimate seismic performance factors.

In modern construction, glue-laminated (glulam) timbers are used to make log-elements. The effects of a first log-foundation configuration, wall slenderness ratios, and vertical pre-compression levels on partial log-walls were examined under monotonic and cyclic loads by Branco et al. [12]. Timber log-wall structures with different configurations (corner intersection types, opening, log-element geometries) were evaluated for their response under in-plane seismic loads in full-scale tests [13,14]. An experimental approach was used to assess the seismic performance of the entire timber log-house. A two-story log-house was built on a triaxial platform shaking table. The fundamental period and damages were determined [15,16].

An advanced FEM of a timber log-wall was created in the ABAQUS commercial software package. Three-dimensional (3D) solid elements and contact interactions were used to model full-scale timber log-wall structures [13]. The results of the 3D-FEM indicated acceptable accuracy compared to the monotonic lateral load test results. Thus, this modeling technique was used to investigate the efficiency of a timber log-wall with steel dovetail reinforcement and to determine the stress distribution on the contact surfaces of historical carpentry corner joints [18,19].

To reduce computational cost, a simplified FEM was proposed for the log-wall structure. Scott et al. [20] used the four-node plane stress element as the log-element and the beam element as the thru-rod and anchor bolt. Nonlinear springs, with properties determined by individual tests, represented the contact interactions between elements. The results of this FEM were consistent with the test data. Branco et al. [12] used nonlinear link elements to represent the shear stiffness of corner joints and log–log interactions. The FEM was calibrated with experimental data from a partial log-wall and used to determine the in-plane stiffness of another log-wall configuration. The unloading and reloading phases of the cyclic response (pinching phenomenon, strength, and stiffness degradations) were not included in the investigation. Bedon et al. [21] used axial and shear hysteresis laws to describe the mechanical response of a single corner joint. This corner joint was considered as a nonlinear spring with three axial spring parameters and 14 shear spring parameters obtained from single carpentry joint test data using So.p.h.i. software [22]. Rigid beam elements were used to represent log-elements. Each rigid beam was connected at its ends by nonlinear springs. The FEM was implemented in ABAQUS and validated with experimental data. Rinaldi et al. [23] applied this modeling technique to study the seismic behavior of light-frame timber structures. The models were implemented in ABAQUS using an external user subroutine and SAP2000 using multilinear plastic (MLP) link elements. The results in SAP2000 seemed less accurate than those in ABAQUS as the MLP-link could not characterize the strength degradation.

Previous research has focused on solid timber log-wall structures. Any timber used as a log-element must meet the quality control criteria of ASTM D 2555 [24] to avoid shrinkage and swelling. Such high-quality wood is rarely found today. Alternative construction materials such as wood polyvinyl chloride (WPVC) composite material have been developed [25,26]. They are light, with better dimensional stability and termite and corrosion resistance. This material was developed into log-elements in the log-house system of Pulngern et al. [27], which evaluated the efficiency of WPVC composite log-walls with five different cross-section designs under compressive load in acoustic and thermal environments. Local and premature failures and uncontrollable deformed shapes were obtained in the compressive load test. The presence of many log-element cross-section flanges reduced the acoustic and thermal resistance capability. Eakintumas et al. [28] designed a new log-element cross-section and used a flat steel bar to strengthen the WPVC composite log-wall. According to the experimental results, the bar helped to control the deformation shape and increased the ductility of the panels. In terms of comfort conditions, the acoustic performance and thermal resistances of a WPVC composite log-house were investigated in a field study and compared to a commercial knockdown house in a previous authors’ work [29]. In 2021, a corner-joint connection configuration and first log-foundation connection were proposed and examined under monotonic and cyclic lateral loading by the authors’ research [30]. The results indicated that an angle bracket first log-foundation connection (LF-AB) and standard half-lapped corner joint (CJ-SHL) were suitable for the WPVC composite log-house.

From the literatures of log-wall behavior under lateral load, numerous research focused on the timber or glue-laminated timber log-element with solid cross section. For WPVC composite log-element with hollow cross-section, the development is in the early stages. The entire behavior of log-wall with WPVC composite material and hollow cross-section of log-element under lateral loads had never been in literature. The effect of composite material and hollow cross-section possibly reflected the different behavior compared to the solid timber log-walls. Furthermore, the appropriate connections, obtained from the authors’ research [30], were used as lateral resisting elements in a full-scale WPVC composite log-wall to evaluate the entire behavior of wall under monotonic and cyclic loading. The in-plane lateral load behavior of the panel with and without additional through-bolts was determined experimentally. The WPVC composite log-wall panel with through-bolts outperformed the log-wall without through-bolts in terms of strength and energy dissipation due to a reduction in wall uplift. A FEM of the WPVC composite log-wall panel with through-bolts was created using beam elements as log-members and multilinear plastic links as connections. The FEM was constructed using SAP2000 software (version 20, CSI), which is a well-known package and has user-friendly interface for structural analysis of seismic problem [31], and validated using the experimental data. Further parametric analyses of wall dimensions and first log-foundation locations were conducted using the validated FEM.

## 2. Experimental Investigation of WPVC Composite Log-Wall

### 2.1. Geometry of Specimens

WPVC composite material was produced by blending wood particles, polyvinyl chloride (PVC) powder, and an additive chemical admixture, supported by V.P. Wood Co., Ltd., Bangkok, Thailand (certified industrial factory). The preparation process consisted of five steps, according to the authors’ previous research [25,26]: (1) drying wood particles (80 weight% of wood particle is rough wood flour (teakwood with average particle size of 700 μm) and 20 weight% of wood particle is fine wood flour (rubberwood with average particle size of 120 μm)) at 80 °C; (2) preparing PVC compound by mixing the suspension and PVC emulsion with a chemical admixture (Pb–Ba-based organic polyfluorene, polyfluorene, high molecular weight complex compatible lubricant, calcium stearate, calcium carbonate, modified chlorinated polyethylene, and polyacrylic); (3) dry-blending of dried wood particles and PVC compound with 1:1 weight ratio; (4) melt-blending of the mixture in a mold using an industrial-scale twin-screw extruder at 180 °C; (5) cooling the hollow WPVC composite members. The average density of the members was 1.283 g·cm^−3^. WPVC composite log-elements consisted of two hollow web members (a plank with 12 holes) and three hollow flange members (a plank with three holes). A cyanoacrylate adhesive was used to glue the webs and flanges, as illustrated in Figure 1.

Two WPVC composite log-walls without through-bolts (W01) and Two WPVC composite log-walls with through-bolts at both ends (W02) were investigated in this research. The layout of the full-scale log-wall is presented in Figure 2. Five main log-members and ten orthogonal log-members were vertically stacked on each other, for a total width of 1488 mm and a total height of 1175 mm. Angle bracket first log-foundation connections (LF-AB) and standard half-lapped corner joints (CJ-SHL) were selected as lateral resisting log-wall elements. CJ-SHL connections consisted of two main and two orthogonal log-elements. The log-element height was trimmed by one-quarter at the upper and lower edges 150 mm from the end, as shown in Figure 2 LF-AB connections consisted of equal angle steel with dimensions of 4 mm × 50 mm × 50 mm, 40 mm in length with four self-tapping No. 7 screws 3.9 mm in diameter and 25.4 mm in length. LF-AB connections connected the first log-element and base support on both sides, 394 mm from each end. Two log-wall panels without through-bolts were subjected to monotonic and cyclic loading as specimen W01. For specimen W02, two M12 through-bolts (12 mm in diameter and 1000 mm in length) were used as an additional lateral resisting element to clamp main log-elements 75 mm from both ends, as shown in Figure 3. Through-bolts were installed at the middle flanges of the first and fifth log-elements. Strain gauges were attached to the through-bolts around the first and second log-element layers to measure tensile strain during monotonic and cyclic loading of specimen W02.

### 2.2. Experimental Setup

The experimental campaign of WPVC composite log-walls were designed by applying the previous research of timber log-wall [12,14]. During the test, the applied lateral loads and lateral displacements of the wall were recorded to generate the load versus displacement curves, which were used to evaluate the behavior and calculate the parameters of the walls.

Log-wall specimens were attached to the steel beam support and restrained to a rigid floor by LF-AB. Five displacement transducers (DT) were installed with a rigid column to measure the horizontal sliding of each log-element (DT A to DT E), as illustrated in Figure 4. DT F was used to measure the distance of the double-action hydraulic jack with 100 kN capacity (C is compressive direction and T is tensile direction). The top log-element was clamped with two steel plates and thread BB bars for load transfer. To reflect the actual behavior of the log-wall, the walls were subjected to two directions of load. The first direction of vertical load represents the applied gravity load while the second direction represents the lateral load from wind force. The double-action hydraulic jack was restrained to the reaction frame to apply the loading protocols (Figure 5) to the top log-element. To apply a gravity load corresponding to the working condition of a one-story log-house with a typical roof span of 4 m, the loads [32] consisted of a roof cover dead load (DL) of 0.049 kN·m^−2^, a roof structure DL of 0.49 kN·m^−2^, devices and ceiling DL of 0.196 kN·m^−2^, WPVC composite log-element self-weight of 0.827 kN·m^−1^ per element, and a roof live load of 0.49 kN·m^−2^. A total load of 2.37 kN, excluding the weight of the instruments, was applied to the specimens. The vertical load was created by thread BB bars and distributed to specimens by loading the head and spreader beams. The wide-flange WF 100 × 100 × 6 × 8 mm was selected as the spreader beam which was used to distribute the vertical load to the wall bearing specimen. This technique was recommended by the previous works of Branco and Araújo [12]. The selected spreader steel beam has high enough of the beam stiffness and the deflection during vertical pre-loading is not occurred. Therefore, the applied forces on the wall were assumed to be uniformly distributed load. Regarding the practical application, the panel can deform or expand (overturning) under lateral loads. To maintain the approximate constant gravity load level during the panel deformed, the steel spring were inserted at one end of each thread BB bars and the gravity load level was monitored by load cell (30 kN capacity). The use of spring allowed the wall specimens to deform (uplift) while the vertical load was maintained, which corresponded to the behavior of building under lateral load, this technique was applied from previous research works [12,14,30]. In this research, the in-plane lateral load was emphasized. Thus, rubber wheels were installed in the middle of the top log-element for lateral support in out-of-plane direction on both sides of the panel (Figure 4).

A monotonic horizontal load was applied to the log-wall with a force control method according to BS EN 26891:1991 [33]. The preliminary specimen was subjected to a pushover test to obtain the estimated failure load (F_est_) before testing by the monotonic protocol, as shown in Figure 5a. The load at 0.4F_est_ was maintained for 30 s, unloaded to 0.1F_est_ with 30 s of constant load, and reloaded at a constant rate until failure. The duration of each cyclic test was in the range 600–780 s (10–13 min) [33]. Therefore, the rate of 0.05 mm·s^−1^ were used.

A cyclic horizontal load with the rate of slip 0.2 mm·s^−1^ was applied according to BS EN 12512:2001 [34], as presented in Figure 5b. An estimated yield slip (V_y, est_) of 5 mm was determined as the average yield slip of CJ-SHL and LF-AB obtained from previous research [30]. The cyclic tests were controlled by lateral displacement. Positive displacement was considered as the compressive direction (C); negative displacement was considered as the tensile direction (T).

### 2.3. Experimental Results and Discussion

The monotonic and cyclic test results for specimen W01 are displayed in Figure 6a. The wall responded to a pushover load with three interaction behaviors: friction, slip to bearing, and wall uplift. The friction between main log-layers occurred at the initial state (H ≈ 0–2 kN). After this phase, the displacement increased as the load was greater than the static friction force (H ≈ 2–3.8 kN). The main log-element slipped with dynamic friction to contact the orthogonal log. In case of timber log-wall [12,14], the timber log-wall show obvious horizontal plateau at the second state of load and displacement curve as contribution of assembly tolerance. The wall began to uplift as the bearings between main- and orthogonal log-elements at corner joints were engaged. At this state, the interlock of corner joints was activated. The wall stiffness decreased as increasing of wall uplift. The log-element layer between the first and second main log-elements began to separate when the lateral load was greater than 3.8 kN (lateral displacement of 5 mm) because the lateral load produced the overturning moment. In this state, load and displacement gradually increased with a constant slope as seen in Figure 6a. The test ended when the log-wall became unstable. A major wall uplift (≈50 mm or 1.32 times of lateral displacement at failure) was observed with a maximum lateral load of 7.15 kN and maximum displacement of 36.8 mm, as shown in Figure 7a. The separation occurred between the first and second main log-elements because the first log-element was restrained to the base support using LF-AB. In contrast, each log-layer was unrestrained in the vertical direction. Tearing of the first log-element was observed, as illustrated in Figure 7b, with a failure shape similar to that of a single-joint experiment in a previous study [30]. This failure was also similar character to the solid timber log-wall [14], which failed by shear brittle failure of the lower part on the notch.

Under cyclic load, Specimen W01 generated an envelope curve in a compressive direction corresponding to the monotonic curve (Figure 6a). The symmetrical shape of the hysteresis curve without cyclic strength degradation was a result of increasing displacement. However, after a displacement of 5 mm, the monotonic load magnitude was less than the cyclic load magnitude (8% lower for 20 mm of displacement) due to the influence of connection relaxation during the reversed load [35]. The panel had the same loading and unloading stiffness; the unloading stiffness decreased as a function of the displacement. The displacement of the unloading path for 10 mm and 20 mm of displacement rapidly returned to overlap the unloading path for a displacement of 5 mm because the wall uplift returned to its normal position. This result confirms that the panel began to uplift after 5 mm of lateral displacement. The increase in lateral displacement at greater displacements influenced wall overturning, as shown in Figure 7a. For repeat cycles, insignificant strength degradation was observed for all displacements (difference less than 1.8%).

The response of wall W02 under monotonic load is shown in Figure 6b. Four phase characteristics are presented: initial state due to friction (H ≈ 0–2 kN), second state due to bearing of corner joints, third state showing stiffness decreasing, and fourth state of specimen failure. The initial state and the beginning of the second state were similar to those for specimen W01; the second state was extended from a load of 2 kN to 8 kN with addition of through-bolts, which helped the log-wall panel to withstand wall uplift between the first and second main log-elements. However, 10 mm of wall uplift (0.38 times of lateral displacement at failure) was observed before failure; it increased with increasing lateral displacement, as shown in Figure 8a. For solid timber log-wall [14], the wall uplift was 0.045 times of lateral displacement at failure, which was lower than that of WPVC composite log-wall due to the fact that the vertical compression of timber log-wall (10 kN) was higher than that of WPVC composite log-wall (2.37 kN).

Load and displacement increased with a decrease in stiffness until the first crack started at the upper screws of the first log-foundation connection. The specimen failed with rupturing of the first log-element, as shown in Figure 8b, with a maximum load and lateral displacement of 9.55 kN and 26.7 mm, respectively. The overturning moment, generated by lateral load, produced the couple force to log-foundation connection. Compressive force transferred to the foundation while tensile force activated the screws and first log-element.

The strain gauge at the through-bolt on the tension side (the same side as the wall uplift) indicated 14.5 μm·m^−1^; the strain gauge at the through-bolt on the compression side indicated 7 μm·m^−1^. Thus, the through-bolts did not directly help to resist the lateral load. In contrast, they helped to increase wall stability by reducing wall overturning, consistent with previous research on solid timber log-walls [9]. Thus, through-bolts were considered as inclinational resisting elements [9].

Wall W02 produced an almost symmetrical hysteresis curve in terms of cyclic response, as shown in Figure 6b. The envelope curve for the cyclic test on the compression side seems stiffer than the monotonic curve in the second state due to log-element relaxation during unloading and reverse loading. Local stresses around the connections were alleviated [35]. Unloading stiffness for this specimen decreased as lateral displacement increased, behaving similarly to W01. The stiffness of the reloading path after unloading at 20 mm of displacement (between point c and point d in Figure 6b) was reduced by wall uplift, similar to the opening and closing gaps in timber framing [36]. This phenomenon is known as pinching, usually found with greater displacement. At displacements of 5 mm and 10 mm, cyclic strength degradations due to repeated loops were less than 1%, whereas the cyclic strength of a loop of 20 mm degraded by 2.5% on the compression side and 4% on the tension side, greater than the results obtained for W01. However, the presence of through-bolts reduced the panel uplift, helping the W02 specimen to generate 26% and 39.5% greater hysteretic energy dissipated per cycle of motion (*E_H_*) than the W01 specimen for displacements of 10 mm and 20 mm, respectively, as shown in Table 1. For a displacement of 5 mm, which had an insignificant effect on wall uplift, the W01 specimen generated a 24.2% greater *E_H_* than the panel with through-bolts.

Moreover, the secant stiffness of cycle (*k_i_*) of the W02 specimen was 26.5%, 41.7%, and 48.2% greater than that of the W01 specimen for displacements of 5 mm, 10 mm, and 20 mm, which directly affected the maximum load (*H_max_*) and elastic strain energy (*E_s_*). The W02 specimen had a 47.8% greater maximum load in the compression side and 56.2% greater maximum load in tension sides than that of W01 specimen for 20 mm of displacement. *E_s_* values for the panel with through-bolts were 14.6%, 42.5% and 51.9% higher than those for the panel without through-bolts. From calculations, equivalent viscous damping ratios (*v_eq_*) of the W01 specimen were 36.4%, 13.0% and 9.2% greater than those of the W02 specimen for 5-mm, 10-mm, and 20-mm displacements, respectively. With the same vibrations, the W01 specimen tended to stop vibrating faster than the W02 specimen. In addition, the panel without a through-bolt had an 8.1% and 10.3% greater recoverable energy (*E_R_*) than the panel with a through-bolt for displacements of 10 mm and 20 mm, respectively, resulting from the rapid return of the unloading path due to wall uplift.

By comparing the hyteretic parameters to solid timber log-wall, it was found that the solid timber log-wall [14] had 53% of *v_eq_* reduction from cycle at *V_est_* to 4*V_est_*, while *v_eq_* of the WPVC composite log-wall (W02) was reduced 29% when displacement level increased from *V_est_* to 4*V_est_.* For log-wall structure, high value of *v_eq_* was obtained at initial state (during static and dynamic frictions), when interlocking of corner joint was activated the value of *v_eq_* was reduced [14]. Solid timber log-wall had higher vertical load, gap torerance, and stiffness due to solid cross-section than WPVC composite log-wall, which were the reason that the *v_eq_* reduction of solid timber log-wall was higher than WPVC composite log-wall. In term of cyclic stiffness at last displacement level, solid timber log-wall (five layers of log-elements) [12] with vertical load of 10.1 kN had 0.836 kN-mm^−1^ of *k_i_*, which was higher than the *k_i_* of WPVC composite log-wall. This was due to the vertical load and the different between solid and hollow cross section.

Comparing W01 and W02, the presence of through-bolts helped to increase the hysteretic parameter values of strength and energy dissipation, except *v_eq_* and *E_R_*, as shown in Table 1. For serviceability, although the panel without a through-bolt presented good damping behavior, the wall uplift directly affected the total structural stability. Thus, the log-wall panel with through-bolts (W02) was selected for further study in nonlinear modeling and parametric investigation.

## 3. Finite Element Model with Nonlinear Hysteretic Link Elements

This research derived nonlinear hysteretic link elements from the cyclic test results for LF-AB and CJ-SHL from previour authors’ work [30] in SAP2000 software. Two nonlinear springs were combined with beam elements to model the entire log-wall. This approach was presented by Rinaldin et al. [23].

### 3.1. Modeling Method

The beam element was used as the WPVC composite log-element. The equivalent log-element cross-section, with a cross-sectional area and moment of inertia equal to those of the actual log-element, was produced and used as a beam element cross-section. Multilinear plastic (MLP) link elements were used as log-foundation connections and corner joints. The envelope curves of single LF-AB and CJ-SHL under cyclic loading from authors’ previous research [30] were used as a multilinear force–deformation definition for link elements. Minimizing the difference in total energy absorption between the actual experiment and the single-joint model was a criterion for selecting the hysteresis type and parameters (accepting a difference of up to 5%). For CJ-SHL, kinematic hysteresis without the requirement of parameters was used because the stiffness degradation and the pinching phenomenon were not observed. The response of LF-AB under cyclic load produced stiffness degradation and a pinching effect. Thus, pivot hysteresis with five parameters was used.

Parameter *α*_1_ was used to control unloading stiffness; the reload stiffness was controlled by parameter *α*_2_. Parameter *β* was used to define the slope of the pinching branch (*β*_1_ for unloading and *β*_2_ for reloading). Parameter *η* defined the elastic stiffness degradation [17]. To control the load–slip path of the LF-AB connection, *α*_1_ = 7, *α*_2_ = 5, *β*_1_ = 0.99, *β*_2_ = 0.95, *η* = 1.0 were defined. Link-element calibrations in the single-joint cyclic test are presented in Figure 9. The difference in total energy absorption between the actual experiment and the single-joint model met the acceptance criteria.

After accurate calibration of nonlinear springs, the FEM of the full-scale WPVC composite log-wall described in Section 2 was constructed using beam elements and MLP links, as shown in Figure 9. Monotonic and cyclic in-plane lateral loads were applied at point A (Figure 10a) as a displacement time history. The left corner joints (leeward side) responded to the applied displacement in the same direction. In contrast, the corner joints on the right side (windward side) reacted to the external displacement in the reverse direction, as displayed in Figure 10b.

### 3.2. Verification of Finite Element Model

Figure 11 shows a comparison of load-displacement relationships obtained from the experiment (EXP) and the FEM. Excellent agreement between the monotonic curve of the experiment and the FEM was found with a lateral load of 8 kN, as shown in Figure 11a For a lateral load greater than 8 kN, the first main log-element started to crack around the log-foundation connection (Figure 8b) through wall uplift. In this research, the equivalent log-foundation MLP links were simplified and considered only as horizontal responses. Thus, the FEM cannot generate the panel behavior after damage has occurred, resulting in differences in the hysteretic curves and parameters at 20 mm of displacement between the experimental and numerical results, as shown in Figure 11b and Table 1. However, most hysteretic parameters with 5 mm and 10 mm of displacement derived from the FEM were sufficiently accurate (within 6.5%). It can be concluded that the FEM can reasonably predict panel behavior with an in-plane lateral load before damage occurs.

## 4. Parametric Studies

From comparison of the experimental and FEM results in Section 3.2, the FEM was extended to further study of a full-scale log-wall panel configuration considering the height of the wall, width of the wall, and log-foundation connection configurations, as illustrated in Figure 12. Further studies were conducted using the validated FEM, which had a wall configuration corresponding to specimen W02. To represent actual behavior, a WPVC composite log-wall with a height of 2.35 m (10 log-elements), a width of 3 m, and a log-foundation connection configuration Case 1 (Figure 12) was defined as the original panel. According to ASCE 7 [37], the allowable story drift was used as a lateral displacement limitation for the model. In this study, the log-house was categorized in seismic risk category I. For other structure types, the limitation of allowable story drift equaled 0.020 of the story height. Thus, the allowable story drift (Δ*_a_*) was 0.047 m (47 mm) when the story height (h) was 2.35 m (Δ*_a_* = 61.1 mm for h = 3.055 m and Δ*_a_* = 70.5 mm for h = 3.525 m).

### 4.1. Wall Dimensions

The height of the wall was studied first. The wall height varied from 2.35 m (10 log-elements) to 3.055 m (13 log-elements) and 3.525 m (15 log-elements). As shown in Figure 13a, the increase in wall height from 2.35 m to 3.525 m resulted the 42.8% lower of initial stiffness, 31.5% lower of final stiffness, and 6.49% lower of maximum lateral load capacity with linear relationship because the wall slenderness increased. The cyclic responses of the wall with different heights are presented in Figure 13b. When the wall height was increased to 3.055 m and 3.525 m, the cyclic stiffness decreased by 27.5% and 39.0%, respectively, with the last displacement level. The number of conner joint increased with increasing of wall height, therefore, the spring in the FEM also increased in series which directly affected to the wall stiffness. Similar results were obtained with solid timber log-wall panels in previous research [12,21]. At the last displacement level, panels with heights of 3.055 m and 3.525 m generated 34% and 78.8% more energy dissipation. Thus, increasing panel height unfavorably affected lateral load capacity, monotonic stiffness, and cyclic stiffness, however, the wall can dissipate higher energy as wall height increase.

In varying the wall width, widths of 3.5 m, 4.5 m, and 6 m were considered. The wall responded to monotonic load, as shown in Figure 14a. The decreases in wall stiffness (26.9% for initial stiffness and 4.61% for final stiffness) and lateral load capacity (13%) were observed when the wall width was increased from 3.5 m to 6 m. This result differs from the result of a typical homogeneous panel, which had lateral load capacity and stiffness increased as wall width increased. In the case of the log-wall structure, the panel was not homogeneous; therefore, increasing the wall width caused the log-element slenderness to increase. The results also differed from the results for solid timber log-walls in previous research [21], which reported that an increase in panel width directly affected friction behavior because the friction area between log-element layers increased. The friction increased the lateral load resistance, cyclic stiffness, and energy dissipation. However, the contribution of pure friction between log-element layers was not considered a lateral resisting element following the Eurocode provisions due to uncertainty [12]. The static friction coefficient of the WPVC composite surface was low due to its smoothness. Thus, friction behavior was excluded from the FEM in this study. For cyclic loading, increased panel width resulted in lower cyclic stiffness (Figure 14b). Compared with the original panel, the cyclic stiffness of panels with widths of 4.5 m and 6 m decreased 7.5% and 14%, respectively. Moreover, a panel with greater width was less able to dissipate energy (11.3% and 24.4% decreases for panels with widths of 4.5 m and 6 m, respectively). It can be concluded that increasing panel width unfavorably influenced lateral load capacity, monotonic and cyclic stiffness, and energy dissipation. However, its influence was less than that of increasing panel height.

### 4.2. First Log-Foundation Connection Locations

Based on the wall uplift observed in the experiment, the number and position of log-foundation connections were considered. Four connection configurations were implemented, as shown in Figure 12. Cases 1–3 emphasized the connection spacing; Case 4 focused on the connection position. The FEM analysis indicated that the spacing of the log-foundation connection did not significantly influence the monotonic and cyclic responses of the WPVC composite log-wall, as illustrated in Figure 15, due to the wall overturning. The maximum wall uplift occurred at the edges of the panel due to the loading directions. Thus, adding log-foundation connections around the center of the panel was ineffective. FEM analysis revealed that moving log-foundation connections from the original position (Case 1) to the edges of the panel (Case 4), which expanded the resisting moment arm, increased 48.1% of initial stiffness, 23.4% of final stiffness, and 16.8% of lateral load capacity and also increased the hysteresis parameters in terms of cyclic stiffness (19.4%) and energy dissipation (15.3%) because installation of log-foundation connection as Case 4 had highest overturning moment arm. These results are consistent with those from a solid timber log-wall in which the connections were transferred from the main log-element to attach the orthogonal log-elements [12]; the experiment indicated that moving the log-foundation connection away from the center of the log-wall produced better performance under cyclic horizontal displacement [12]. A summary of parametric studies is presented in Table 2.

## 5. Conclusions

In this research, LF-AB and CJ-SHL connections were used in a full-scale WPVC composite log-wall. The in-plane lateral load behaviors of the panel with and without additional through-bolts were determined experimentally. A FEM of the WPVC composite log-wall panel with through-bolts was created using beam elements as log-members and multilinear plastic links as connections. The FEM was constructed using SAP2000 software, and validated using the experimental data. Further parametric analyses of wall dimensions and first log-foundation locations were conducted using the validated FEM. The key findings of this study are presented as follows:The WPVC composite log-wall without through-bolts (W01) generated a maximum monotonic load of 7.15 kN and a maximum displacement of 36.8 mm. The panel became unstable with significant wall uplift and failed by tearing the first main log-element;The panel with through-bolts (W02) responded to monotonic load with a maximum load and lateral displacement of 9.55 kN and 26.7 mm, respectively. The experiment also indicated that through-bolts did not directly resist the lateral load. However, they helped to increase wall stability by reducing wall overturning. Through-bolts were considered as inclinational resisting elements;Under cyclic loading, through-bolts increased hysteretic parameter values in terms of strength and energy dissipation (48.2% for secant stiffness of cycle and 39.5% for hysteretic energy at the last displacement level), except equivalent viscous damping and recoverable energy. Although the panel without through-bolts exhibited good damping behavior, for serviceability conditions, the overturning of without through-bolts directly affected the overall structural stability. Thus, the log-wall panel with through-bolts (type 2) was selected for further study;Comparison of experimental data and analysis results from a simplified two-dimensional FEM indicated that a FEM with beam elements as log-members and multilinear plastic links as connections can be used to predict WPVC composite log-wall behavior under an in-plane lateral load before damage occurs;In the parametric investigations, increasing panel height and width unfavorably affected lateral load capacity, monotonic stiffness, and cyclic stiffness. Energy dissipation increased with a panel height increase while it decreased with an increase in panel width. The panel had a higher lateral load capacity, monotonic stiffness, and cyclic stiffness. There is energy dissipation as the resisting moment arm increases.

## Figures and Tables

**Figure 1 polymers-14-04673-f001:**
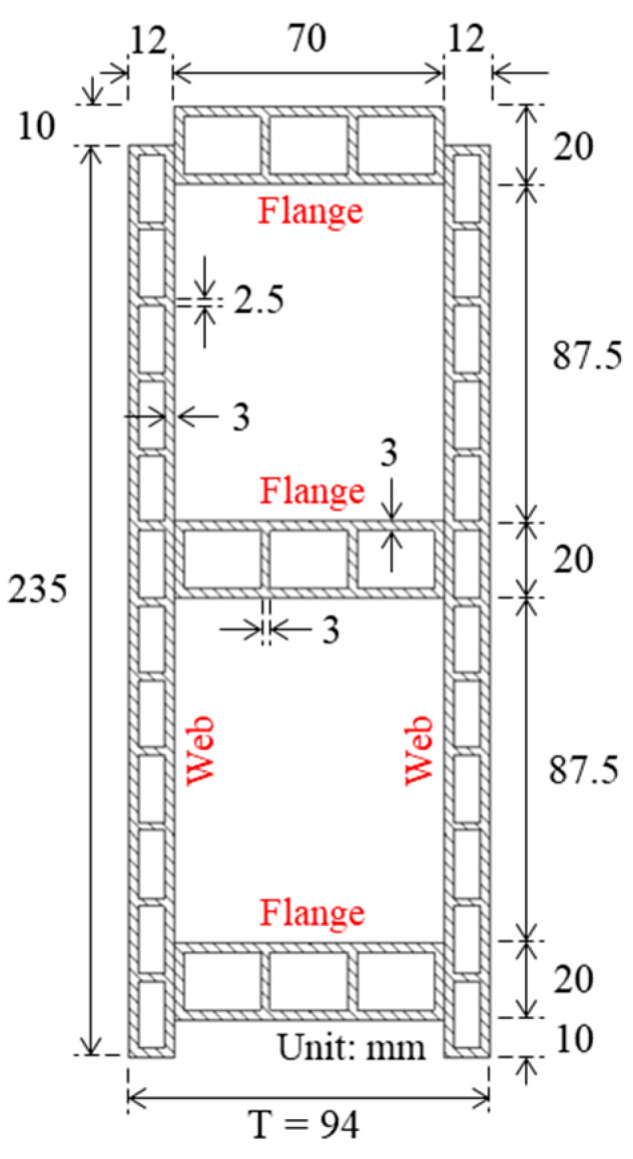
Dimensions of typical WPVC composite log-element cross-section (T is the thickness of log-element).

**Figure 2 polymers-14-04673-f002:**
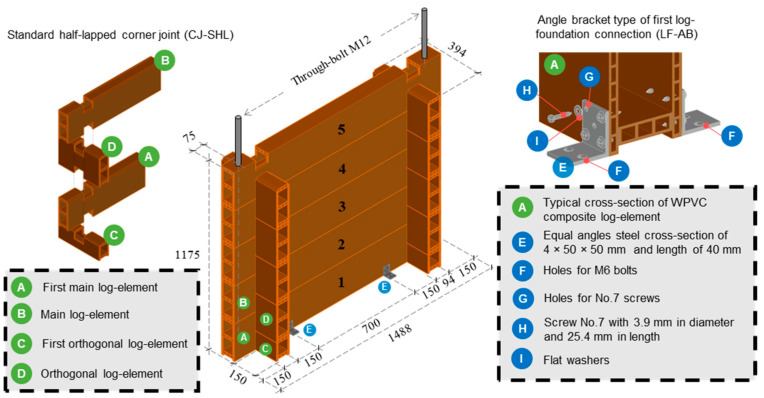
Layout of full-scale WPVC composite log-wall specimens W01 and W02.

**Figure 3 polymers-14-04673-f003:**
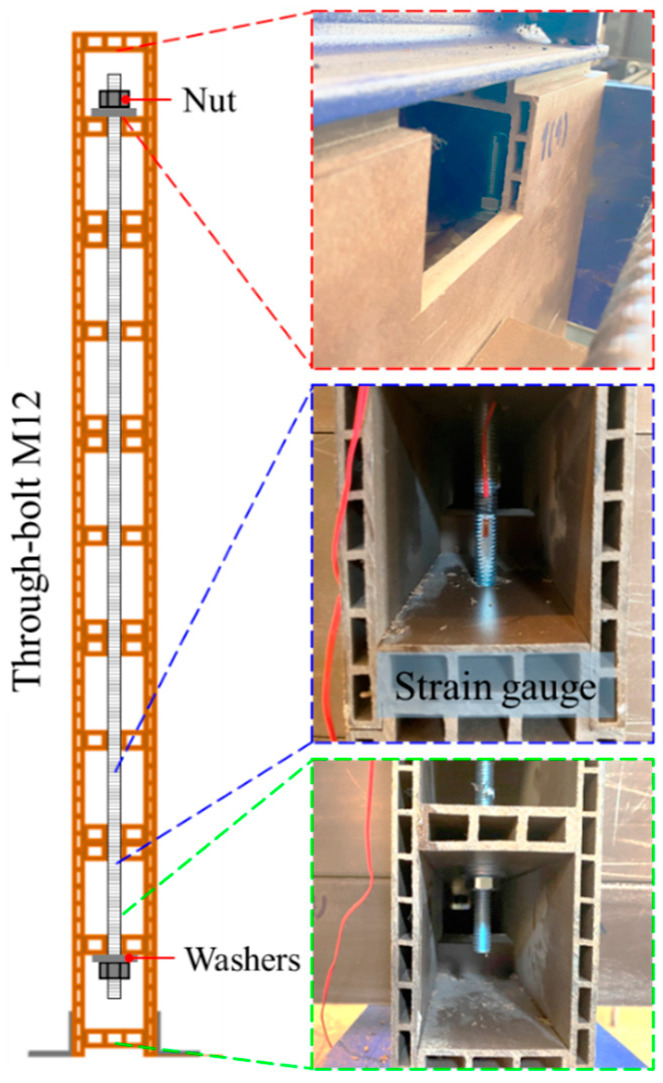
Detail of through-bolt connection for specimen W02.

**Figure 4 polymers-14-04673-f004:**
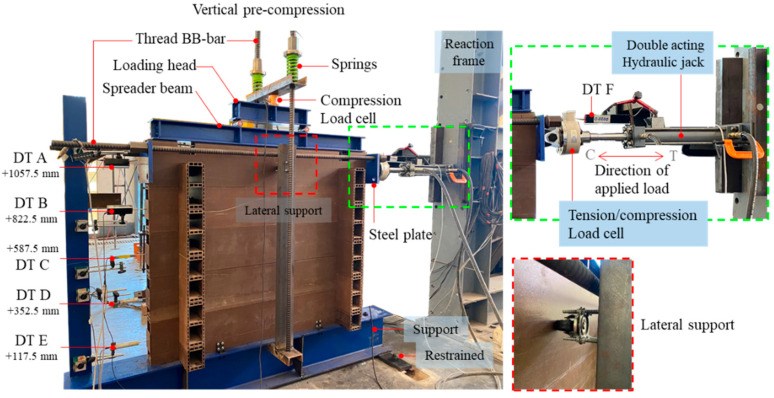
In-plane lateral load experimental setup (instrument positions and lateral support) performed on WPVC composite log-wall (C is compressive direction and T is tensile direction).

**Figure 5 polymers-14-04673-f005:**
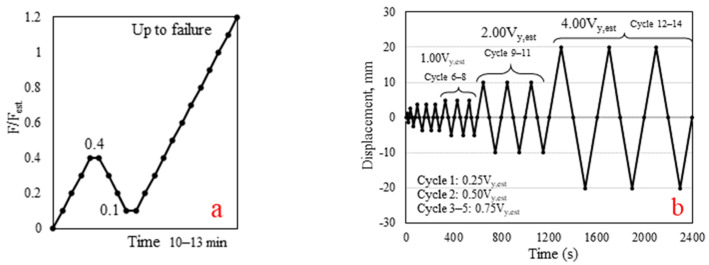
Horizontal in-plane loading protocol for (**a**) monotonic; (**b**) cyclic loading.

**Figure 6 polymers-14-04673-f006:**
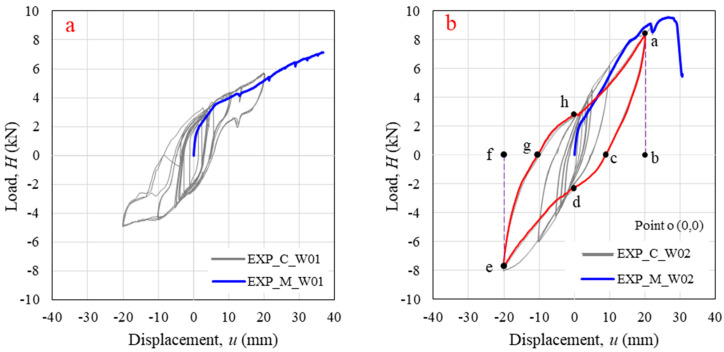
Horizontal in-plane lateral load versus displacement relationship (at DT A) under monotonic (EXP_M) and cyclic (EXP_C) loading for (**a**) Specimen W01; (**b**) Specimen W02.

**Figure 7 polymers-14-04673-f007:**
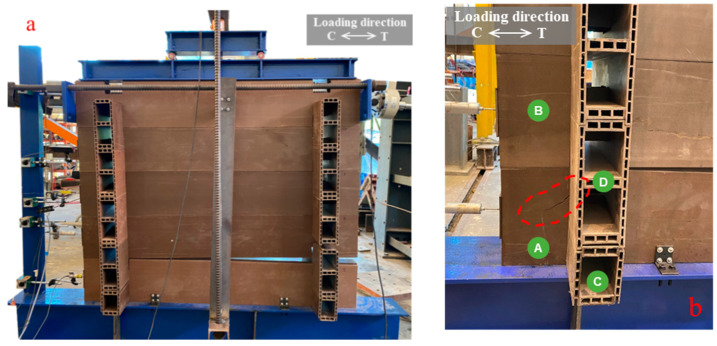
WPVC composite log-wall specimen (W01) under compressive loading: (**a**) Overall deformed shape; (**b**) Tearing of first main log-member (log A).

**Figure 8 polymers-14-04673-f008:**
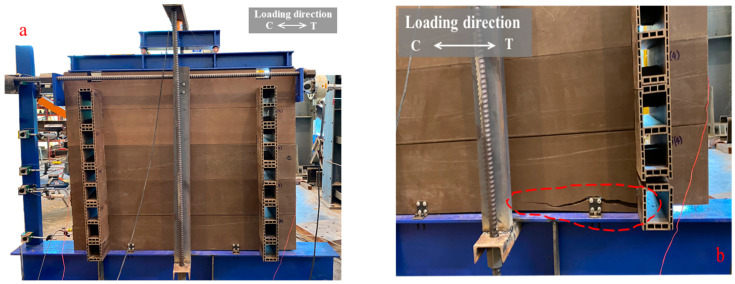
WPVC composite log-wall specimen (W01) under compressive loading: (**a**) Overall deformed shape; (**b**) Separation between first main log-member and support.

**Figure 9 polymers-14-04673-f009:**
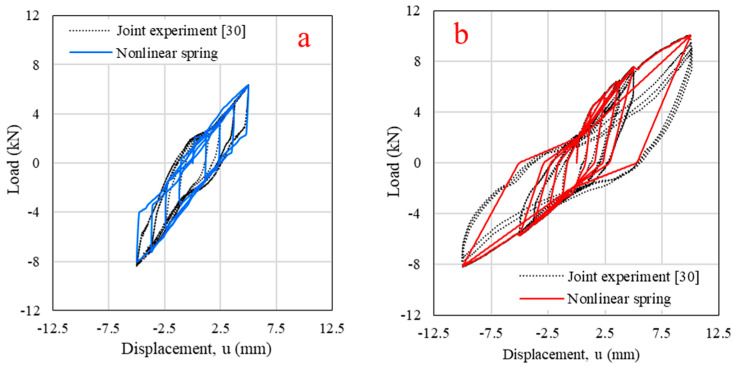
Nonlinear spring calibration with previous experiment for (**a**) Standard half-lapped corner joint (CJ-SHL); (**b**) Angle bracket first log-foundation connection (LF-AB).

**Figure 10 polymers-14-04673-f010:**
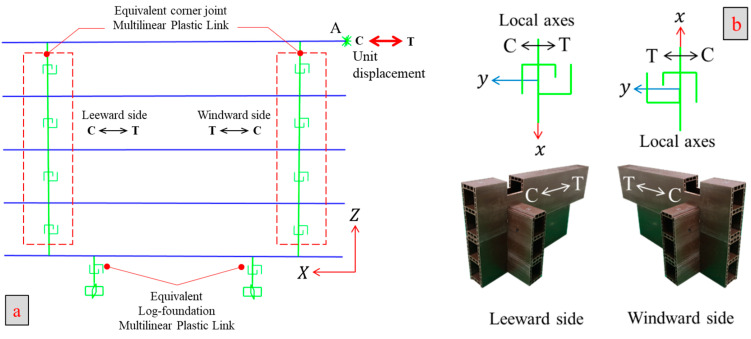
Finite element modeling schematic for WPVC composite log-wall: (**a**) Overall view; (**b**) Details of corner joints on leeward and windward sides.

**Figure 11 polymers-14-04673-f011:**
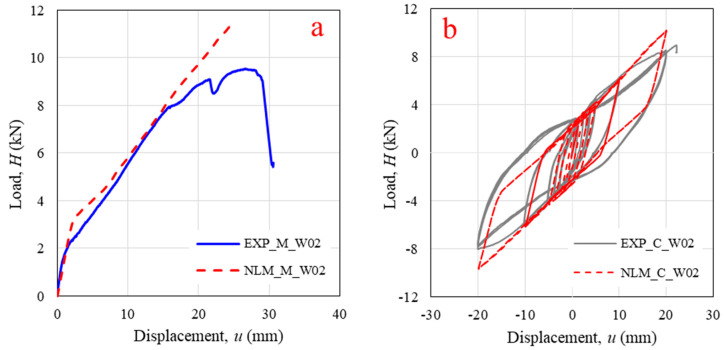
Comparison of experimental (EXP) and finite element model (FEM) results for specimen W02 under (**a**) monotonic loading; (**b**) cyclic loading.

**Figure 12 polymers-14-04673-f012:**
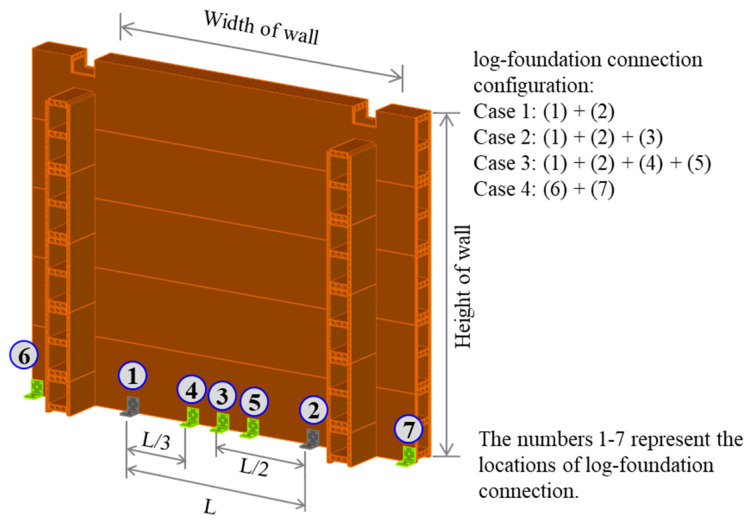
Schematic of WPVC composite log-wall for parametric studies.

**Figure 13 polymers-14-04673-f013:**
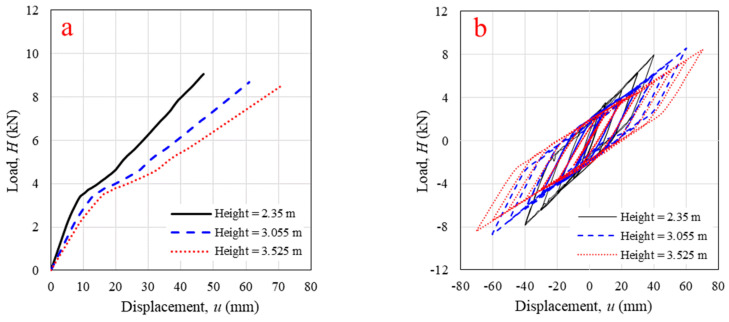
Horizontal in-plane lateral load versus displacement from validated numerical modeling with different log-wall heights under (**a**) monotonic loading; (**b**) cyclic loading.

**Figure 14 polymers-14-04673-f014:**
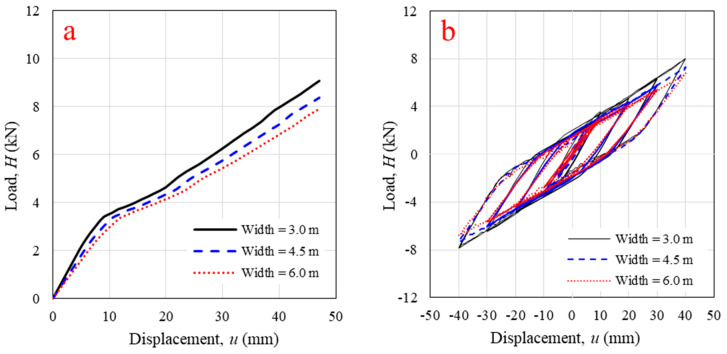
Horizontal in-plane lateral load versus displacement from validated numerical modeling with different log-wall widths under (**a**) monotonic loading; (**b**) cyclic loading.

**Figure 15 polymers-14-04673-f015:**
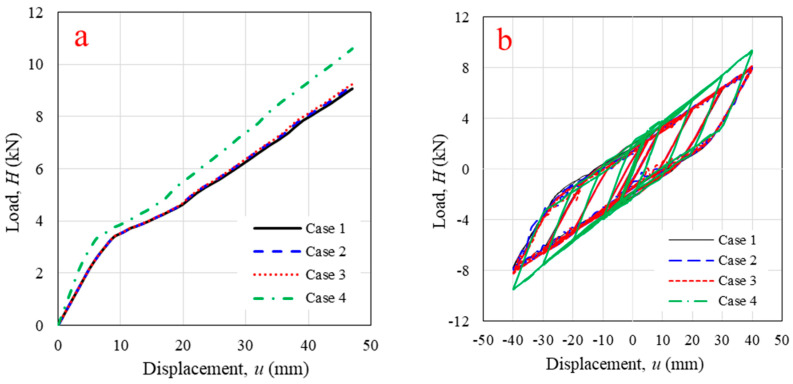
Horizontal in-plane lateral load versus displacement from validated numerical modeling with different log-foundation connection configurations under (**a**) monotonic loading; (**b**) cyclic loading.

**Table 1 polymers-14-04673-t001:** Experimental and numerical hysteretic parameters for specimens W01 and W02, and FEM panel.

Wall		Load Type	*H_max_* at 20 mm (kN)		Cyclic Parameters	*V_est_* (Cycle 8th)	2*V_est_* (Cycle 11th)	4*V_est_* (Cycle 14th)
			Compression	Tension				
W01	EXP	Monotonic	5.20	-				
		Cyclic	5.65	−4.93	*k_i_*	0.68	0.43	0.27
					*E_H_*	38.0	59.3	119
					*E_S_*	19.2	43.5	106
					*E_R_*	4.49	22.8	65.3
					*v_eq_*	31.5	21.7	17.9
W02	EXP	Monotonic	8.88	-				
		Cyclic	8.35	−7.70	*k_i_*	0.86	0.61	0.40
					*E_H_*	30.6	74.7	166
					*E_S_*	22.0	62.0	161
					*E_R_*	5.75	21.1	59.2
					*v_eq_*	23.1	19.2	16.4
W02	FEM	Monotonic	9.72	-				
		Cyclic	10.2	−9.62	*k_i_*	0.85	0.61	0.50
					*E_H_*	32.6	72.1	159
					*E_S_*	21.2	61.3	198
					*E_R_*	9.38	21.5	93.1
					*v_eq_*	24.5	18.7	12.8

*k_i_* = (*H_com,i_* − *H_ten,i_*) / (*u_com,i_* − *u_ten,i_*) (kN·mm^−1^), *E_H_* = *Area_acdegh_* (kN-mm), *E_R_* = *Area_abc_* + *Area_efg_* (kN-mm), *E_S_* = *Area_oab_* + *Area_oef_* (kN-mm), *v_eq_* = *E_H_*/(2*π*·*E_S_*) (%), subscript characters of *Area* are the points in Figure 6b.

**Table 2 polymers-14-04673-t002:** Summary of parametric studies.

Parameters	Lateral Load Capacity	Monotonic Stiffness	Cyclic Stiffness	Energy Dissipation
Increasing panel height from 2.35 m to 3.525 m	−6.49%	−42.8%	−39.0%	+78.8%
Increasing panel width from 3.5 m to 6.0 m	−13.0%	−26.9%	−4.61%	−24.4%
Increasing resisting moment arm	+16.8%	+48.1%	+19.4%	+15.3%

Where: + stand for increasing, and − stand for decreasing.

## Data Availability

The data presented in this study are available on request from the corresponding author.

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
