# Peer review of "Finite Element Modeling and Experimental Investigation for Wood/PVC Composites Log-Walls under In-Plane Lateral Load"

_polymers, 2022, doi:10.3390/polym14214673_

Round 1

Reviewer 1 Report (Previous Reviewer 1)

The amended manuscript should be accepted.

Author Response

Reviewer 2 Report (Previous Reviewer 2)

The paper can be accepted without any further changes.

Author Response

Reviewer 3 Report (Previous Reviewer 3)

Please read the attachment. Thank you.

Author Response

This manuscript is a resubmission of an earlier submission. The following is a list of the peer review reports and author responses from that submission.

Round 1

Reviewer 1 Report

This manuscript was not well written. The novelty of this work should be clarified. STAT and experimental design should be carried out. SD should be added to all results. For discussion, deep scientific reasons for all results should be provided. Also, please compare it with other previous works. Therefore, this manuscript cannot be published in this Journal. 

Reviewer 2 Report

This research paper present on in-plane lateral load behavior of the panel with and without additional through-bolts of WPVC. The topic relate very much with the structural engineered wood based and it was a novel idea in this research field. Few comments regarding the content stated below.

P3 L142 - Please mention detail regarding SAP2000 software for reader understanding.

P3 L147 - Author(s) should mention the species used in this research even though they were using mixed wood particles.

P6 L205-214 - Regarding the monotonic and cyclic protocol, how long the sample was taken to be tested? Please stated the time duration in the Figure 5.

Reviewer 3 Report

Please find the attached file in this email.
